# The Generalizability of HERO across 15 Nations: Positive Psychological Capital (PsyCap) beyond the US and Other WEIRD Countries

**DOI:** 10.3390/ijerph17249432

**Published:** 2020-12-16

**Authors:** Stewart I. Donaldson, Lawrence B. Chan, Jennifer Villalobos, Christopher L. Chen

**Affiliations:** Division of Behavioral & Organizational Sciences, Claremont Graduate University, University in Claremont, Claremont, CA 91711, USA; lawrence.chan@cgu.edu (L.B.C.); jennifer.villalobos@cgu.edu (J.V.); christopher.chen@cgu.edu (C.L.C.)

**Keywords:** positive psychological capital (PsyCap), HERO model, positive organizational psychology, cross-cultural positive organizational behavior, work performance

## Abstract

Recent meta-analyses of positive organizational psychology interventions (POPIs) suggest that interventions that target and improve hope, efficacy, resilience, and optimism (HERO) can be highly effective at improving well-being and positive functioning at work. However, many studies to date have been conducted with samples from the US and other Western, educated, industrialized, rich, and democratic (WEIRD) societies, which raise the concern about the generalizability of theory-driven POPIs. The aim of this study was to examine if the underlying mechanism of one of the most successful POPIs to date, positive psychological capital (PsyCap) based on the HERO model, predicts positive functioning at work across diverse geographical regions and cultures. Using Qualtrics Panel data collected from 3860 employees across 15 nations (Australia, Brazil, China, France, Germany, Great Britain, India, Ireland, Italy, Japan, Netherlands, New Zealand, Philippines, South Africa, and the United States), we found that PsyCap is strongly associated with workplace proactivity, proficiency, adaptivity, and overall work performance across all 15 nations. The results suggest that efforts to develop PsyCap may be effective across national cultures and could be a robust approach for enhancing positive functioning in the global workplace.

## 1. Introduction

It is a great honor to be invited to contribute to this Special Issue entitled: “Challenges in Positive Organizational Psychology”. After much reflection, we have decided to attempt to make a contribution toward understanding a challenge that is often raised in this rapidly developing area of inquiry: do positive organizational psychology findings generalize beyond the US and similar Western nations [1,2]?

Donaldson and Ko [3] defined positive organizational psychology POP “as the scientific study of positive subjective experiences and traits in the workplace and positive organizations, and its application to improve the effectiveness and quality of life in organizations.” At that time, they identified two related multidisciplinary streams of scholarship and research—positive organizational behavior [4] and positive organizational scholarship [5], which greatly contributed to the understanding of positive organizational psychology topics such as positive leadership, positive organizational development and change, positive psychological capital (PsyCap), organizational virtuousness and ethics, well-being at work, work engagement, flow at work, and more recently, positive organizational psychology interventions. Furthermore, Salanova, Llorens, and Martínez [6] underscored the importance of positive organizations providing an environment and culture where workers can develop, as well amplify and enhance well-being and optimal functioning. Recent systematic reviews have highlighted the empirical evidence now supporting the value of this approach to organizational scholarship, research, and intervention [3,7,8].

There are also growing number of successful examples of rigorously tested positive organizational psychology interventions (POPIs) designed to improve well-being and optimal functioning in the workplace [9]. A recent meta-analysis of the strongest studies to date found that on average POPIs have had small-to-moderate positive effects across both desirable and undesirable work outcomes (e.g., job stress), including well-being, engagement, leader member exchange, organization-based self-esteem, workplace trust, forgiveness, prosocial behavior, leadership, and calling [10]. Further analyses have found the following type of POPIs appear to be most effective when strong measurement and evaluation designs have been used to determine their effectiveness [11]:Positive psychological capital interventionsJob crafting interventionsEmployee strengths interventionsEmployee gratitude interventionsEmployee well-being interventions

While the evidence supporting the effectiveness of POPIs and positive psychology interventions (PPIs) is encouraging [9], the concern remains that many rigorous tests of PPIs and POPIs have been carried out in Western, educated, industrialized, rich, and democratic (WEIRD) countries [12,13,14,15]. This leaves researchers in the field of positive organizational psychology unsure about the generalizability of efforts to improve well-being and positive functioning at work by implementing POPIs.

The challenge of knowing whether or not the findings from positive organizational psychology research to date generalize across national contexts and cultures and beyond WEIRD nations will be explored empirically in this paper. We focus on the mechanism responsible for one the most successful POPIs—positive psychological capital (PsyCap). The purpose of our investigation is to examine if PsyCap predicts positive functioning at work across a wide range of nations.

Luthans, Avolio, Avey, and Norman [16] developed the HERO model also known as positive psychological capital (PsyCap) as a higher order construct that represents four positive subjective experiences at work: hope (redirecting paths to work goals), resilience (ability to bounce back at work), self-efficacy (confidence in one’s ability to succeed at work), and optimism (positive attributions about the future of work). PsyCap is a “state-like” construct that is optimal for POPIs due to its flexibility and openness to development. In a recent systematic review of the empirical literature, PsyCap POPIs were found to be a robust approach for increasing well-being and performance in organizations [15]. However, most of the empirical studies of PsyCap POPIs were conducted in WEIRD countries (87% of the reviewed studies), and the authors highly recommended future empirical work is needed to extend the literature.

The present study was designed to determine whether PsyCap predicts the performance of employees from a wide variety of cultural backgrounds and national origins. We investigated the utility of PsyCap as a central predictor of positive functioning at work across a sample of 15 countries comprising of a range of first-world and developing countries: Australia, Brazil, China, France, Germany, Great Britain, India, Ireland, Italy, Japan, Netherlands, New Zealand, Philippines, South Africa, and the United States. Griffin, Neal, and Parker [17] proposed a framework and validated measures to study work role performance in uncertain and interdependent contexts. This framework yielded valid measures of adaptivity, proficiency, and proactivity, which can also be combined to assess overall work role performance across diverse organizations [18].

Given the uncertain and interdependent nature of work contexts assessed in our study, we used the work role performance scale to measure performance across nations [17]. Internal reliability statistics conducted from prior validation studies were high across both the total scale and within each sub dimension, with Cronbach’s α > 0.76 [17]. The measure also considers cross-functional performance and incorporates consideration of team-level and organization-level performance. Considering all 15 countries and using the three sub-dimensions of the work role performance measure, we hypothesized (see Figure 1):

**Hypothesis** **1** **(H1).**
*PsyCap will predict high adaptive performance across employees from diverse nations.*


**Hypothesis** **2** **(H2).**
*PsyCap will predict high proactivity across employees from diverse nations.*


**Hypothesis** **3** **(H3).**
*PsyCap will predict high proficiency across employees from diverse nations.*


**Hypothesis** **4** **(H4).**
*PsyCap will predict high overall work performance (aggregate measure of proficiency, proactivity, and adaptivity) across employees from diverse nations.*


## 2. Methods

### 2.1. Participants and Design

Secondary data analyses were conducted on a subset of a larger dataset collected by a multinational professional services firm with the goal of identifying behavioral factors for success in the future workforce. The firm granted authors permission to utilize this de-identified dataset.

Study sample. Pre-screened respondents were invited to join the study on a voluntary basis and received compensation for their involvement through Qualtrics Panels. Participants in the dataset were representative of the working population in their respective countries. Participant names and other forms of identification were anonymized in the received dataset. We chose to use this dataset because it thoroughly encompasses participants from multiple countries across the world and is comprehensively representative of various cultures. In order to retain the accurate representation of the workforce for each country, all participants (other than those removed during data cleaning) were included in our analyses.

Data cleaning. During the data cleaning process, 76 missing cases were removed (*n* = 3784). Consistent with prior theory [19,20] univariate and multivariate outliers were removed to minimize their disproportionate influence: 89 univariate outliers (*n* = 3695) were removed that were outside ± 3 standard deviations [21], and 81 additional cases were subsequently removed using Mahalanobis’ cutoff distance for multivariate outliers (±3 standard deviations), resulting in a final *n* = 3614 [22,23].

Sample characteristics. Table 1 illustrates descriptive statistics. Following data cleaning procedures, final participants included were *n* = 3614 (1855 men (51.32%) and 1758 women (48.64%)) employees sampled across 15 countries. Participants’ median age was 37 years old. Participants comprised workers from Australia (*n* = 216, 6.00%), Brazil (*n* = 210, 5.81%), China (*n* = 220, 6.10%), France (*n* = 216, 5.97%), Germany (*n* = 220, 6.10%), Great Britain (*n* = 216, 5.97%), India (*n* = 377, 10.42%), Ireland (*n* = 216, 5.97%), Italy (*n* = 211, 5.89%), Japan (*n* = 218, 6.02%), Netherlands (*n* = 217, 6.00%), New Zealand (*n* = 211, 5.84%), Philippines (*n* = 223, 6.16%), South Africa (*n* = 221, 6.11%), and the United States (*n* = 422, 11.67%). Levels of education across our final sample were representative of most working professionals: high school diploma = 288 (7.96%), associate = 528 (14.62%), bachelor = 1926 (53.29%), master’s degree = 734 (20.32%), doctorate = 138 (3.82%).

### 2.2. Measures

Participants responded to all study questions using a 7-point Likert scale (from 1 = “strongly disagree” to 7 = “strongly agree”). Measures used for this study were part of a larger survey (51 items). The initial data sought to identify psychological factors that are crucial for employee success in the future workforce. Items that were not pertinent to the current analyses were omitted. Study instruments were translated by a professional translation services firm so that all respondents completed the study in their native languages.

Positive Psychological Capital. PsyCap was measured using an adapted version of the 12-item short form Psychological Capital Questionnaire-12 (PCQ-12) [16]. The measure assesses the level of self-efficacy, hope, resilience, and optimism among respondents as an aggregate behavioral construct. Because this subset of data originated from a larger study of several scales, the researchers adapted the initial PCQ-12 to minimize survey length and overall subject burden while completing the study. As such, researchers also validated the scale using statistical procedures outlined below, in which four of the items with the lowest factor loadings were omitted, resulting in an adapted 8-item version of the scale. Examples of sample items are “I feel confident presenting information to a group of colleagues” and “I am optimistic about what will happen to me in the future as it pertains to work”. The response set is a seven-point Likert scale (1 = “strongly disagree” to 7 = “strongly agree”). A high score indicates higher PsyCap, whereas a low score indicates low levels of PsyCap. Prior validation studies suggest strong evidence for psychometric validity (α = 0.89) [16].

Work Role Performance. Work role performance indicators of proactivity, adaptivity, and proficiency were measured using the 27-item Work Role Performance Scale [17]. The Work Role Performance Scale assesses individual proclivity for adaptivity, proactivity, and proficiency as indicators of performance. Examples of sample items are “I carry out the core parts of my job well” and “I adapt well to changes in core tasks.” For all items, participants were asked to rate how often they had carried out the behavior on a scale ranging from (1 = “strongly disagree” to 7 = “strongly agree”). A high score indicates higher work role performance, whereas a low score indicates low levels of work role performance. Prior validation studies suggest strong evidence for psychometric validity (α = 0.86) [17]. Our own alpha analysis from the study sample confirmed these findings (α = 0.97).

Control Variables. We assigned certain demographic variables as control variables that may account for variance in our proposed relationship between PsyCap and work role performance. Following recommendations by Johnson et al. [24], we inspected zero-order correlations to identify the variables that shared variance with the predictor but not the criterion. We also controlled for variables that could lead to an unnecessary reduction in statistical power, as well as an increase in Type I errors. Subsequently, three control variables were included in the analysis: age, gender, and level of education. There is some research on PsyCap to suggest that age is a significant predictor of PsyCap [25] and ultimately performance, while gender and level of education have greater variability of scores on employee performance [4]. Moreover, previous findings purport that similar results may be found in predicting work role performance [17].

### 2.3. Analytic Strategy

PsyCap 8-item CFA. Prior to testing our study hypotheses, we first ran a series of CFA analyses to confirm the strength of the 8-item adapted version of the PCQ-12. In accordance with recommendations of Hu and Bentler [26] and Kyriazos [27], the root-mean-square error of approximation (RMSEA), comparative fit index (CFI), and the standardized root-mean-square residual (SRMR) values were examined, and confirmed model fit, RMSEA ≤ 0.06, CFI ≥ 0.95, SRMR ≤ 0.08.

Regression analyses. We analyzed all scale data with a simple linear regression model using R Statistical computing and graphics software [28,29]. Early research has established that several factors may influence our study variables and should be accounted for [24,30]. There is some evidence to suggest, for example, that individuals with more formal education have higher status jobs and therefore are more likely to develop greater degrees of PsyCap in their professional capacities [24]; thus, driving enhanced work performance. Moreover, the same argument can be applied to an individual’s age, as older aged individuals experience increased career opportunities and longer tenures in their work environments, they also are more likely to have developed greater degrees of resilience, hope, optimism, and efficacy at work [24]. Although there is some evidence to suggest that there are no significant gender differences in PsyCap [31], this research has not been investigated cross-culturally. Given the findings from the aforementioned studies, we thus accounted for education level (1 = associate degree, 2 = bachelor’s degree, 3 = master’s degree, 4 = doctorate degree, 5 = high school diploma or equivalent), age (birth year), gender (0 = male, 1 = female, 2 = other), and country. Given our large dataset across 15 countries, we also ran an additional moderation analysis to assess whether the relationship between PsyCap and work role performance varied as a function of the variable “country”. Predictor variables were centered before creating the interaction terms [32]. Significance and variance estimates were calculated and are reported in the results section.

## 3. Results

### 3.1. Descriptive Statistics and Model Fit

Means, standard deviations, and correlations between all variables are presented in Table 1. Gender variables were coded as male = 0 and female = 1 for comparison purposes to other study variables. Similarly, the variable for education degree was coded as associate = 1, bachelor = 2, master = 3, doctorate = 4, high school (or equivalent) = 5. Although control variables (age, education, and gender) correlate with our outcome variables for work role performance (proficiency, proactivity, and adaptivity) and PsyCap, these correlations were modest; the largest of which is between the level of education and PsyCap (r = 0.10, *p* < 0.001). The strongest correlations exist between our primary predictor variable PsyCap, and the work role performance variables of proficiency, adaptivity, and proactivity.

Given the high bivariate correlations (i.e., all significant at a level below 0.01) among our whole sample predictor (PsyCap) and outcome (proficiency, adaptivity, and proactivity) variables, we ran additional analyses to assess whether these strong correlations exist across countries (Table 2). To highlight similarities in variance estimates across the countries, countries were also grouped according to WEIRD versus non-WEIRD categories, as defined by Hendriks et al. [1]. We found that strong correlations between PsyCap and our performance variables remained consistent throughout each country.

### 3.2. Hypothesis Testing

Hypotheses 1–4. We entered the predictor variables into simple linear hierarchical regression analyses in these steps: (1) the predictor variable PsyCap plus the control variables of age, gender, and education with the outcome variable of adaptivity; (2) the predictor variable PsyCap plus the control variables of age, gender, and education with the outcome variable of proactivity; (3) the predictor variable PsyCap plus the control variables of age, gender, and education with the outcome variable of proficiency; and (4) the predictor variable PsyCap plus the control variables of age, gender, and education with the outcome variable of overall work performance.

Given the growing literature on cross-cultural differences in personality and performance management [33,34], we deemed it important to subset our sample by the 15 countries to better ascertain the extent of differences in the performance sub-dimensions of adaptivity, proactivity, proficiency, and overall performance.

Hypothesis 1 addresses individual level PsyCap as a predictor of workplace performance variable of adaptivity. Supporting this hypothesis, regression coefficients for PsyCap were significant in predicting adaptivity across all 15 countries (Table 3).

Hypothesis 2 addresses individual level PsyCap as a predictor of workplace performance variable of proactivity. Supporting this hypothesis, regression coefficients for PsyCap were significant in predicting proactivity across all 15 countries (Table 3).

Hypothesis 3 addresses individual level PsyCap as a predictor of workplace performance variable of proficiency. Supporting this hypothesis, regression coefficients for PsyCap were significant in predicting proficiency across all 15 countries (Table 3).

Hypothesis 4 addresses individual level PsyCap as a predictor of overall work performance composite variable, including three sub-dimensions of proactivity, adaptivity, and proficiency, consistent with the Griffin and colleagues’ [17] Work Role Performance measure. Supporting this hypothesis, regression coefficients for PsyCap were significant in predicting overall workplace performance across all 15 countries (Table 3).

Consistent with previous research and our study hypotheses, these findings further support that PsyCap is a robust predictor of performance, as defined by our measure of work role performance, including individual performance indicators of proactivity, adaptivity, and proficiency. As expected, age, gender, and education remained insignificant across all three dimensions in most countries, except for education and age in a small sample of countries. However, even in those limited cases, variance estimates attributed to education and age are minuscule and practically insignificant. The moderation term for the variable of “country” was also a non-significant predictor in our regression model, suggesting that geographical region does not moderate the relationship between individual PsyCap and work role performance. In addition, the insignificance of the effect of country justifies our decision not to compare grouping by the category of WEIRD versus non-WEIRD countries. Table 3 reports variance results for performance variables by country at a significance level of *p* < 0.01.

## 4. Discussion

PsyCap as a higher order construct has been shown to be an important predictor of employee workplace performance in modern organizations [10,11,15], while its individual components of hope, self-efficacy, resilience, and optimism have also been shown to be antecedents to a plethora of work outcomes, including high performance, work engagement, team cohesion, and return on investment [3,18,35,36,37,38,39]. One practical strength of this empirical evidence is that it clearly supports that PsyCap can be developed and can be used to design human resource development and positive organizational psychology interventions (POPIs) [10,11,15]).

Much of the previous research on PsyCap POPIs has shown that they can be highly effective in WEIRD workforces [10,11]. However, as organizations have become increasingly more global and diverse, there have been concerns about whether PsyCap predicts performance across diverse cultures and nations [15,40]. The findings of this study provide new empirical evidence that support PsyCap is a robust construct that can be applied across many countries and diverse workplace settings. More specifically, PsyCap was found to predict work adaptivity, proactivity, proficiency, and overall work performance across 15 diverse nations. Even after controlling for age, education level, and gender, PsyCap accounted for the majority of variance across all four measures of work role performance. One important practical implication of our findings is that POPIs, which are successful at developing PsyCap, are likely to be effective at enhancing positive functioning at work in diverse and global workplaces and support efforts to promote diversity and inclusivity [37,41].

### 4.1. Strengths and Limitations

One strength of this study is the robust dataset that allowed us to examine PsyCap across diverse cultures and nations. Very few studies have systematically studied PsyCap with such a diverse sample of employees. Given the prioritization of diversity and inclusivity in the modern workplace, shedding new insights on this popular construct from a broad and well-represented sample expands what we know about best practices for human resource managers and organizational leaders in this context.

It is important to outline some of the possible limitations of our analyses. Given that our analyses relied on a cross sectional correlational survey design, future research may consider using multiple measures of these constructs (e.g., employee, co-worker, and manager reports) and longitudinal designs to understand if the relationships hold beyond mono-method bias and over time [42,43]. Future research to replicate these findings using experimental designs testing the efficacy of POPIs based on PsyCap theory and research in non-Western countries could expand our understanding of this approach to improving work performance across diverse nations. Though ideal in some ways, these designs are more invasive than the systematic measurement study we conducted and may not be as plausible for cross cultural research in many large corporate or professional services organizations.

### 4.2. Future Directions

Replicating these findings across a new sample of countries could further support our conclusion that PsyCap is likely to be a robust predictor of performance across cultures. For example, future research examining specific job types, career levels, and popular industry contexts within each of the countries assessed will also help us to better understand the boundary conditions of PsyCap. Drawing on the recent trends of positive organizational psychology research, future research examining PsyCap may seek to engage specific working populations within each of the countries included to establish whether specific contexts influence performance, such as healthcare, social service, and education [44,45,46].

This research suggests that PsyCap is a reliable predictor of adaptive work roles and behaviors that human resource managers can develop in a rapidly changing environment. Managing the modern gender and culturally heterogenous workforce has pushed organizational leaders to explore new talent management practices so that this new generation of workers are developed and assessed in a fair and inclusive manner. Not only have diversity and inclusion talent management practices become top priorities of senior leaders, but considerations of diversity and inclusivity are also now part of mainstream corporate branding and organizational purpose statements [47]. As such, PsyCap interventions that meet a standard of inclusivity hold great promise for enhancing positive functioning in the global workforce. Whether current PsyCap intervention frameworks meet this standard is an area for future research.

## 5. Conclusions

Our findings provide new evidence that developing PsyCap based on the HERO model could be a robust and inclusive global human resource strategy for enhancing positive functioning at work. The practical implications of these findings are important because human resource development leaders have increasingly called on organizational researchers to help show the value of organizational cultures and workforces that embody diversity and inclusivity. It is our hope that future efforts will focus on designing and evaluating PsyCap and positive organizational psychology interventions in general for diverse employees across the global workplace.

## Figures and Tables

**Figure 1 ijerph-17-09432-f001:**
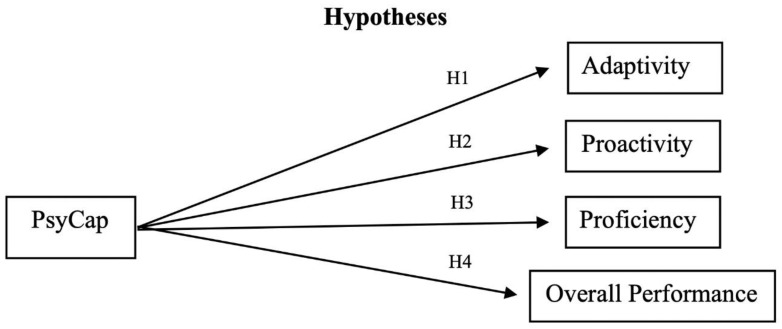
Hypotheses.

**Table 1 ijerph-17-09432-t001:** Means, standard deviations, and the intercorrelations among study variables.

Variable	*M*	*SD*						
Age	36.44	10.19						
Gender	0.49	0.50	−0.06 **					
			[−0.10, −0.03]					
Education	1.97	0.91	−0.08 **	−0.04 *				
			[−0.12, −0.05]	[−0.07, −0.01]				
PsyCap	5.43	0.90	0.02	−0.05 **	0.10 **			
			[−0.02, 0.05]	[−0.08, −0.02]	[0.07, 0.13]			
Proficiency	5.79	0.82	0.05 **	0.06 **	0.06 **	0.73 **		
			[0.02, 0.08]	[0.03, 0.10]	[0.02, 0.09]	[0.72, 0.75]		
Adaptivity	5.63	0.86	0.03	0.03 *	0.07 **	0.75 **	0.80 **	
			[−0.01, 0.06]	[0.00, 0.07]	[0.04, 0.11]	[0.73, 0.76]	[0.79, 0.81]	
Proactivity	5.48	0.94	−0.00	−0.02	0.09 **	0.70 **	0.69 **	0.76 **
			[−0.04, 0.03]	[−0.05, 0.01]	[0.06, 0.13]	[0.68, 0.71]	[0.67, 0.71]	[0.74, 0.77]

Note. *M* and *SD* are used to represent mean and standard deviation, respectively. Values in square brackets indicate the 95% confidence interval for each correlation. * indicates *p* < 0.05. ** indicates *p* < 0.01.

**Table 2 ijerph-17-09432-t002:** Zero order correlations between PsyCap and job performance.

Variable		Country	*n*	Adaptivity	Proactivity	Proficiency	Overall Performance
PsyCap	WEIRD	Australia	216	0.77 *	0.68 *	0.78 *	0.80 *
		France	214	0.77 *	0.71 *	0.75 *	0.80 *
		UK	215	0.76 *	0.70 *	0.72 *	0.80 *
		Germany	215	0.73 *	0.72 *	0.74 *	0.79 *
		Ireland	216	0.69 *	0.67 *	0.69 *	0.76 *
		Italy	213	0.81 *	0.76 *	0.79 *	0.85 *
		Netherlands	214	0.76 *	0.69 *	0.68 *	0.79 *
		New Zealand	211	0.75 *	0.67 *	0.76 *	0.81 *
		South Africa	219	0.68 *	0.65 *	0.66 *	0.75 *
		USA	426	0.69 *	0.55 *	0.64 *	0.71 *
	Non-WEIRD	Brazil	211	0.74 *	0.72 *	0.76 *	0.80 *
		China	222	0.84 *	0.79 *	0.85 *	0.88 *
		India	381	0.76 *	0.71 *	0.72 *	0.78 *
		Japan	216	0.84 *	0.83 *	0.78 *	0.87 *

Note. *n* = 3614. * indicates *p* < 0.01.

**Table 3 ijerph-17-09432-t003:** Percent variance contributed by PsyCap to job performance by country.

Variable		Country	*n*	Adaptivity	Proactivity	Proficiency	Overall Performance
PsyCap	WEIRD	Australia	216	0.612 *	0.476 *	0.630 *	0.661 *
		France	214	0.551 *	0.514 *	0.730 *	0.664 *
		UK	215	0.586 *	0.498 *	0.565 *	0.653 *
		Germany	215	0.549 *	0.526 *	0.565 *	0.642 *
		Ireland	216	0.676 *	0.463 *	0.500 *	0.736 *
		Italy	213	0.676 *	0.590 *	0.631 *	0.736 *
		Netherlands	214	0.580 *	0.509 *	0.516 *	0.659 *
		New Zealand	211	0.580 *	0.457 *	0.589 *	0.666 *
		Philippines	225	0.560 *	0.544 *	0.589 *	0.627 *
		South Africa	219	0.466 *	0.437 *	0.441 *	0.570 *
		USA	426	0.485 *	0.310 *	0.466 *	0.519 *
	Non-WEIRD	Brazil	211	0.551 *	0.519 *	0.576 *	0.649 *
		China	222	0.551 *	0.634 *	0.730 *	0.778 *
		India	381	0.483 *	0.509 *	0.550 *	0.631 *
		Japan	216	0.707 *	0.698 *	0.631 *	0.736 *

Note. *n* = 3614. * indicates *p* < 0.01.

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
