# Peer review of "The Generalizability of HERO across 15 Nations: Positive Psychological Capital (PsyCap) beyond the US and Other WEIRD Countries"

_ijerph, 2020, doi:10.3390/ijerph17249432_

Round 1

Reviewer 1 Report

Thank you for the opportunity to review the manuscript entitled “Understanding the Generalizability of Positive Organizational Psychology Interventions: A Focus on Positive Psychological Capital (PsyCap)” (ijerph-999298).

The authors report a cross-sectional correlational study on the association of positive psychological capital (PsyCap) with work performance in 3,860 employees from 15 countries. In all countries, PsyCap was significantly related to work performance. This is the core of this study. Therefore, the manuscript promises much more than it can deliver, starting by the title, the abstract, and the key words. No conclusions about the generalizability of positive organizational psychology interventions, the underlying mechanisms of such interventions etc. can be done. In addition, it is doubtful whether this manuscript fits the scope of this journal.

Other points are:

Why did the authors remove outliers (+/- 3 SD) in questionnaire data?

Abbreviations should be introduced when they are first mentioned.

The sentence in line 274 – 276 should be deleted.

Words that imply causality (e.g., predictor) should be avoided in such a cross sectional study

Why was the PsyCap scale adapted?

Why were the countries not categorized into WEIRD countries vs. not?

Author Response

Thank you for the opportunity to review the manuscript entitled “Understanding the Generalizability of Positive Organizational Psychology Interventions: A Focus on Positive Psychological Capital (PsyCap)” (ijerph-999298).

The authors report a cross-sectional correlational study on the association of positive psychological capital (PsyCap) with work performance in 3,860 employees from 15 countries. In all countries, PsyCap was significantly related to work performance. This is the core of this study. Therefore, the manuscript promises much more than it can deliver, starting by the title, the abstract, and the key words. No conclusions about the generalizability of positive organizational psychology interventions, the underlying mechanisms of such interventions etc. can be done. In addition, it is doubtful whether this manuscript fits the scope of this journal.

We have reframed the paper to set more accurate expectations for our data and findings.  We have revised the title, abstract, and introduction to accomplish this.  We have made it clear we were invited to contribute the paper to a special issue to address the surprising comment “, it is doubtful whether this manuscript fits the scope of this journal.”

Other points are:

Why did the authors remove outliers (+/- 3 SD) in questionnaire data?

Thank you for this underscoring this issue. We revised this section to indicate that the decision to remove univariate and multivariate outliers is supported by theory as a method to remove their influence in the dataset.

Abbreviations should be introduced when they are first mentioned.

We have revised the manuscript to incorporate this suggestion.

The sentence in line 274 – 276 should be deleted.

We have revised the manuscript to delete this sentence.

Words that imply causality (e.g., predictor) should be avoided in such a cross sectional study

We have revised the manuscript so that it is more clear we are not implying our findings are causal in nature.

Why was the PsyCap scale adapted?

We have revised the manuscript to include the rationale for adapting the PsyCap, including the validation measures we included in the analytic strategy.

Why were the countries not categorized into WEIRD countries vs. not?

Thank you for this suggestion. Countries are now categorized into WEIRD countries and Non-WEIRD countries in Tables 2 and 3. Variance estimates for both remain consistent between the WEIRD and Non-WEIRD countries.

Reviewer 2 Report

The claim that PsyCap has only been studied in WEIRD countries is overstated.  Furthermore the sample includes NZ, Australia, Ireland and Germany.  As such, the introduction needs to be re worked away from the claim of moving beyond WEIRD countries/samples, towards framing this study as a cross-national study. Alternatively, the authors could spend more time discussing the similarities or differences in outcomes over WEIRD v other countries.

The analysis is limited.  Some discussion of why the similarities /differences between countries was found (and whether there was any statistical significance in these) would add to the viability of the project.  For example, do those countries that are not WEIRD face additional challenges that PsyCap equips these employees to deal with?  Regardless there is a lack of depth around analysis and discussion.

However, it is a good sound review.  The theoretical advancement (beyond WEIRD) maybe overstated and as such needs to be modified.  Future studies may also examine between country outcomes beyond performance, and examine the differing antecedents to PsyCap.

Author Response

The claim that PsyCap has only been studied in WEIRD countries is overstated.  Furthermore the sample includes NZ, Australia, Ireland and Germany.  As such, the introduction needs to be re worked away from the claim of moving beyond WEIRD countries/samples, towards framing this study as a cross-national study. Alternatively, the authors could spend more time discussing the similarities or differences in outcomes over WEIRD v other countries.

The analysis is limited.  Some discussion of why the similarities /differences between countries was found (and whether there was any statistical significance in these) would add to the viability of the project.  For example, do those countries that are not WEIRD face additional challenges that PsyCap equips these employees to deal with?  Regardless there is a lack of depth around analysis and discussion.

However, it is a good sound review.  The theoretical advancement (beyond WEIRD) maybe overstated and as such needs to be modified.  Future studies may also examine between country outcomes beyond performance, and examine the differing antecedents to PsyCap.

We have reframed the paper as described above to address these helpful suggestions for improvement.

Reviewer 3 Report

This paper tested the relationships between psychological capital and work performance measures. Overall, the research methodology is valid, which was based on a large dataset crossing 15 countries. 

My comments are only related to how the authors interpreted the results based on Table 1 to Table 3. 

  • For table 1, what does 1, 2, 3, 4, 5, 6 represent?What are the range numbers in the brackets? Gender and Education, please use the percent, instead of mean and sd. In the text part, please also give more description of the results. 
  • For table 2, please give more description, lines 204-206 are not clear at all. What is the trend of the data?
  • For table 3, again, please give more description of the results. Simply saying the results are significant is not enough. 

Author Response

This paper tested the relationships between psychological capital and work performance measures. Overall, the research methodology is valid, which was based on a large dataset crossing 15 countries. 

My comments are only related to how the authors interpreted the results based on Table 1 to Table 3. 

  • For table 1, what does 1, 2, 3, 4, 5, 6 represent?What are the range numbers in the brackets? Gender and Education, please use the percent, instead of mean and sd. In the text part, please also give more description of the results. 

Thank you for this feedback. We have removed the bracket numbers from Table 2. As stated in the table notes, the range numbers in brackets indicate the 95% confidence interval for each correlation. We revised the text to clarify the rationale for using means and SD for Gender and Education, which allowed us to compare these categorical variables with continuous variables.

  • For table 2, please give more description, lines 204-206 are not clear at all. What is the trend of the data?
  • Thank you for this feedback. We revised this section so that it is clearer and portrays the trend of the data.
  • For table 3, again, please give more description of the results. Simply saying the results are significant is not enough. 

Thank you for this helpful feedback. We revised this section in an attempt to communicate more clearly about our results.

Round 2

Reviewer 1 Report

I thank the authors for their thorough revision of the manuscript. And I would like to apologize as I was not aware of the fact that this was an invitation for a special issue.

Some small issues however remained: In the hypothesis, the authors still use the word “predict”. Table 1 is not readable.